# The Arabic Version of the Cohen Perceived Stress Scale: Factorial Validity and Measurement Invariance

**DOI:** 10.3390/brainsci11040419

**Published:** 2021-03-26

**Authors:** Amira Mohammed Ali, Amin Omar Hendawy, Ohoud Ahmad, Haleama Al Sabbah, Linda Smail, Hiroshi Kunugi

**Affiliations:** 1National Center of Neurology and Psychiatry, Department of Mental Disorder Research, National Institute of Neuroscience, Tokyo 187-0031, Japan; hkunugi@ncnp.go.jp; 2Department of Psychiatric Nursing and Mental Health, Faculty of Nursing, Alexandria University, Alexandria 21527, Egypt; 3Department of Biological Production, Tokyo University of Agriculture and Technology, Tokyo 183-8509, Japan; amin.hendawy@gmail.com; 4Department of Animal and Poultry Production, Faculty of Agriculture, Damanhour University, Damanhour 22516, Egypt; 5Department of Public Health Nutrition, Zayed University, Dubai 19282, United Arab Emirates; 201510933@zu.ac.ae (O.A.); haleemah.AlSabah@zu.ac.ae (H.A.S.); 6Department of Mathematics & Statistics, Zayed University, Dubai 19282, United Arab Emirates; linda.smail@zu.ac.ae; 7Department of Psychiatry, Teikyo University School of Medicine, Tokyo 173-8605, Japan

**Keywords:** Cohen Perceived Stress Scale, depression, life satisfaction, university students, exploratory factor analysis, confirmatory factor analysis, measurement invariance, psychometric, validity, reliability, United Arab Emirates

## Abstract

University students experience high levels of stress due to university transition, academic commitments, and financial matters. Higher stress perceptions along with limited coping resources endanger mental health for a considerable number of students and may ruin their performance. The current study evaluated the psychometric properties of the Cohen Perceived Stress Scale (10 items), PSS-10, in a sample of 379 female Emeriti students. Exploratory factor analysis resulted in two factors with eigenvalues of 3.88 and 1.19, which explained 60.6% of the variance. Confirmatory factor analysis revealed good model fits of two correlated factors (Comparative Fit Index (CFI) = 0.962, Tucker–Lewis Index (TLI) = 0.950, standardized root-mean-square residual (SRMR) = 0.0479, and root mean-square error of approximation (RMSEA) = 0.067). Internal consistency of the PSS-10 and its positive and negative subscales was acceptable (coefficient α = 0.67, 0.79, and 0.86, respectively). Multigroup analysis revealed that the PSS-10 holds invariance across different groups of age, marital status, and financial status (average monthly expenditure). Convergent and concurrent validity tests signify the importance of considering scores of subscales of the PSS-10 along with its total score.

## 1. Introduction

Stress is an inevitable aspect of everyday life [1]. However, research shows that university students experience higher levels of stress than the general population in the United Kingdom, United States (US), Canada, Australia, Sweden, and many parts of the world [2,3]. Nevertheless, differences in the prevalence rates of stress among university students are associated with variations in cultural background, study design, year of study, and measures used to assess stress [3].

According to their chronological age, most university students live a distinct developmental period that represents a transition between adolescence and young adulthood—described by some psychologists as a developmental crisis, which entails enormous stress and challenges [2,4]. University students have a high load of social, financial, and academic stress, which involve concerns about adapting to a new environment, academic demands, self-funding of education, part-time job requirements or overworking, and future employment. Such high charge of stress can significantly alter students’ health and performance [2,3,5]. However, the transactional model of stress emphasizes that stress perception plays a major role in determining psychological and physiological stress responses elicited under conditions in which environmental demand is perceived to outweigh the ability to cope with the demand [1].

Individual resources, including the ability to properly appraise stress and the capacity to adapt to difficult situations, can remarkably help students bounce back from events, and even advance their personal development [3,5]. On the other side, high stress appraisal may alter body homeostasis and induce excessive physiological distress via a mechanism that involves persistent activation of the sympathetic nervous system and dysregulation of the hypothalamic–pituitary–adrenal axis [6,7]. Students with high stress perception, especially those below the age of 26 years, tend to demonstrate a range of negative outcomes and pathological states such as depression, suicidal ideation, lower subjective wellbeing, and abuse of alcohol and illicit drugs [2,3,4,8]. Collective clinical data obtained from 340 university and college counseling centers in the US over 5 academic years (2010–2015) denoted significantly rising trends for self-reported family distress, academic distress, generalized anxiety, depression, and social anxiety: the last three problems exhibit large effect sizes [9].

A recent meta-analytic review on preventive interventions for emotional negativity and stress among university students reported that mismatching symptoms with adequate interventions limits the effects of these programs. In other words, proper screening and identification of students in real need for treatment are necessary for preventive approaches to yield desirable effects [4]. In this respect, soundly calibrated (valid and reliable) psychometric measures of stress are necessary to identify proper treatment targets and accurately monitor changes in stress levels during and after the provision of mental health support [2].

Developed in 1983 by Cohen and colleagues as a measure of individuals’ appraisal of life events as stressful, the Perceived Stress Scale (PSS) comprised 14 Likert-type items. The PSS-14 was originally intended as a measure of a single factor, global perceived stress. Subsequent dimensionality investigations uncovered that items of the PSS-14 cover two factors described as distress/helplessness and self-efficacy/coping [2,7,10]. A study assessed the structure of the PSS-14 in 2387 US residents—that sample included both genders along with different groups of age, ethnicities, and level of income. The analysis resulted in a 10-item scale, which portrays the same two factors as the PSS-14 but enjoys a better internal measurement integrity [7,10]. Nonetheless, one study reported that the PSS-10 exhibits a 3-factor structure (distress, coping, and emotional reactivity) in healthy Australians and a 2-factor structure (only distress and coping) in healthy Australians with chronic work stress [1].

The PSS-10 is a non-specific, context-free measure of distress, which can be applied in a variety of groups (e.g., students, athletes, drug users, older adults, pregnant and postpartum women, and patients with systemic lupus erythematosus) across a range of settings [7,10,11]. Simplicity and brevity of the PSS-10 make it appealing since it is easy to complete and allows simultaneous inclusion of other measures, a less possible option when the test battery is long [2]. Nonetheless, application, interpretation, and usefulness of the PSS-10 may be potentially limited by inconsistent reports on its dimensionality as well as by invariances across different groups, e.g., gender, age, marital status, and taking part in sport activities [2,7,12,13]. Thus, maximum usability of data derived from the PSS-10 can be achieved through appreciation of its psychometric properties within specific groups [2].

Until the current moment, applicability of the PSS-10 to Arab students has not been considered in a single study. More, the available Arabic version, which is tested among pregnant and postpartum women, has been evaluated via exploratory factor analysis (EFA) [10]. Although EFA has long been one of the most widely used statistical methods in psychological research, EFA has received considerable criticism. In addition to being purely data-oriented with less capacity to provide meaningful insights into data, EPA offers less options for improving measurement purity when items cross-load or have low loadings on the corresponding factors [14]. In EFA, the number of factors to retain is decided based on the criterion of an eigenvalue above 1, which results in overestimation of the number of latent constructs [15]. In this regard, decisions based on poor choices may alter results in a drastic way [14].

On the other side, confirmatory factor analysis (CFA) employs a range of fit indices to examine data fit to a priori theoretical assumption [2]. In other words, it examines if specific models of interest properly fit the data [16]. Virtually, a good fitting model is signified by a non-significant chi-square (χ^2^) statistic, which evaluates both the covariance matrix and the sample. Nevertheless, χ^2^ value tends to increase with large sample size, and it may reflect trivial discrepancies between the model and the data in good fitting models rather than poor model fit [2,16]. Therefore, model data fit in absence of item correlations can be best estimated by the Comparative Fit Index (CFI), the Tucker–Lewis Index (TLI), the root mean-square error of approximation (RMSEA), and the standardized root-mean-square residual (SRMR). Adequate and good fits, in order, are indicated by CFI and TLI values above 0.90 and 0.95, along with RMSEA and RMR values less than 0.06 and 0.08 [2,15]. CFA also examines mutual and causal relationships among latent variables and measured variables [16].

Given the above background, the current study aimed to evaluate the dimensional structure of the PSS-10 in a sample of Arab female university students through the use of EFA and CFA. It also examined the possibility of PSS-10 invariance as a function of age, marital status, and monthly expenditure. Internal consistency reliability of the PSS-10 was evaluated by alpha coefficient, while convergent validity was assessed by item-total correlations. Convergent validity is a measure of construct validity. It assesses if a scale corresponds to another measure that evaluates the same construct [17]. A form of convergent validity is discriminant validity, which identifies the degree of convergence of items of a scale as reflected by correlations of items with domains of the scale and the total score of the scale. Although item-total correlations represent an internal consistency test, they also convey convergent validity because they reflect on ability levels of the participants—the extent to which items may differentiate subjects who score higher (high item-total correlations) on the trait under investigation from those who score lower (low item-total correlations), which is also known as discrimination index [18,19,20]. Concurrent validity is a type of criterion validity that assesses if a scale corresponds to other related but different measures. Concurrent validity was assessed by evaluating the association of the PSS-10 with depression and life satisfaction. In this respect, we hypothesized that the PSS-10 positively correlates with depression and negatively correlates with life satisfaction (Figure 1).

## 2. Materials and Methods

### 2.1. Participants and Procedure

This study employed a cross-sectional design with a sample that comprises 10% of students enrolled in different schools of Zayed University during Spring 2018. Female participants (*N* = 385) were recruited from randomly selected classes—teachers holding these classes were contacted in order to facilitate data collection. During class time, the participants were handed a consent form and the test battery. Of all approached students (388), only 3 students refused to participate. However, a few items had one or two missing responses, which were excluded from the analysis, resulting in a final sample of 379 participants. The study was approved by the Research and Ethics Committee at Zayed University, the United Arab Emirates (UAE).

### 2.2. Instruments

Data were collected via a self-administered questionnaire, which consisted of several parts, as described below.

#### 2.2.1. Sociodemographic Data

The first part of the questionnaire comprised questions about education and socio-demographic data such as specialty, age, city of residence, average monthly expenditure, marital status, etc.

#### 2.2.2. Cohen’s Perceived Stress Scale-10 Items (PSS-10)

The PSS-10 prompts subjects to rate the degree to which they felt that life situations during the past month were unpredictable, uncontrollable, stressful, and overwhelming. Responses come on a 5-point response scale (0 = never, 1 = almost never, 2 = sometimes, 3 = fairly often, 4 = very often). All items of the PSS-10 are negatively worded except for Items 4, 5, 7, and 8, which are positively worded. A total score of the PPS-10 is obtained by summing up all item scores after reverse-coding positively worded items. Higher scores denote higher levels of perceived stress. This Arabic version was obtained from the formal website of the PSS (https://www.cmu.edu/dietrich/psychology/stress-immunity-disease-lab/scales/index.html, accessed on 24 March 2021), with no information available regarding its psychometric properties.

#### 2.2.3. Beck Depression Inventory-II (BDI-II)

For the purpose of concurrent validity testing, participants were administered BDI-II, a common measure of depression. BDI-II consists of 21 items and responses are rated on a 4-point scale that ranges from 0 to 3. Higher scores indicate higher levels of depression. In this study, Item 9, which inquiries about suicidal thoughts, was not addressed to the students as recommended by the ethical committee. The reliability of this 20-item BDI-II in this sample was good (coefficient α = 0.88).

#### 2.2.4. Life Satisfaction

Life satisfaction was measured by a single item, which asked participants to rate their overall level of life satisfaction in the last few days on a scale that ranges from 1 (completely dissatisfied) to 10 (completely satisfied).

### 2.3. Statistical Analysis

To identify the number of factors covered by the PSS-10, we conducted EFA with maximum-likelihood extraction and direct Oblimin rotation to let items freely load on corresponding factors without enforcing any constraints. EFA revealed a 2-factor structure of the PSS-10: one comprising negatively worded items and the other comprising positively worded items. Based on the outcome of EFA, the 2-factor structure of the PSS-10 was evaluated through CFA. Model fit indices (CFI, TLI, RMSEA, and SRMR) in a model involving absence of correlations among items and factors suggested a need for further improvements. Thus, modification indices were checked, and another model with two correlated factors was examined.

Age and monthly expenditure were examined for normality of distribution. Accordingly, participants were categorized into two groups based on the median (20 years and 20,000 Emirati dirhams (ED), respectively). Marital status was categorized into two groups (single and married) because few participants were divorced (who were coded as single) or engaged in a formal relation (who were coded as married). To determine whether the attained factorial solution of the PSS-10 applies to different groups and that items of the scale depict experienced distress independent of other characteristics (e.g., age) [18], measurement invariance of the PSS-10 was examined across groups of age, monthly expenditure, and marital status. Model fit was examined in every group separately. Then, measurement invariance was tested through multi-group CFA by examining the fit of four models relative to the most fitting unconstrained model: (1) configural invariance, which examines the extent to which the overall model generally fits similarly across groups, i.e., the model expresses the same number of factors representing the data in both groups, (2) metric invariance is tested by constraining factor loadings to be equal between groups and examining the difference between the unconstrained and constrained model, (3) strong/scalar invariance by constraining the intercepts of the 10 items of the PSS-10 to be the same between the two groups, and (4) strict invariance by constraining residuals to be equal between the two groups [2]. Invariance across subgroups is indicated by significant changes in model fit, mainly ΔCFI and ΔRMSEA should not exceed 0.02 and 0.015 respectively, because CFI and RMSEA are less dependent on sample size [2,7]. On the other hand, χ^2^ may signify change in model fit across groups; however, it is largely dependent on sample size [7].

Internal consistency reliability of the PSS-10 and its subscales was examined by Cronbach’s alpha, while convergent validity was evaluated by item-total correlations. Construct validity was established by Pearson correlations between perceived stress, its subscales (distress, coping), depression, and life satisfaction. Analysis was conducted in SPSS version 22 and Amos version 24. Significance was considered at a probability less than 0.05 in two-tailed tests.

## 3. Results

### 3.1. Characteristics of the Participants

Participants of this study (mean age = 20.2 ± 3.4, range = 17–29 years) were predominantly single (88.1%) with an average monthly expenditure of 2356 ± 1811.8 ED (range = 30–10,000 ED). Most participants (75.5%) were residents of the Emirate of Dubai. Students studying at the first, second, and third years represented 23.4%, 23.9%, and 19.6% of the sample, respectively.

### 3.2. Factorial Structure of the Arabic PSS-10

#### 3.2.1. Exploratory Factor Analysis

The Kaiser–Meyer–Olkin (KMO) test indicated that the participant-to-item ratio was proper for an EFA test (KMO = 0.848). Likewise, Bartlett’s test denoted appropriateness of the sample size for this analysis (χ^2^(45) = 1422.358, *p* < 0.001). EFA using Oblimen rotation revealed that items of the PSS-10 cover two clearly distinct factors (distress and coping) with eigenvalues > 1 (Figure 2), accounting for 38.8% and 21.9% of the variance, respectively. Table 1 plots item loadings to their corresponding factors—negatively worded items loaded on factor 1 (distress), while positively worded items loaded on factor 2 (coping).

#### 3.2.2. Confirmatory Factor Analysis

Two possible models were tested for good fit. The first model tests the two-factor structure of the PSS-10 under complete absence of correlation between factors and errors. This model had an acceptable fit (Table 2). However, modification indices signified considerable improvements in all model fit indices when both factors correlated, as in model 2, suggesting that model 2, the two-correlated factors, represent the best fit for these data (Figure 3). There were moderate–strong correlations (r = 0.60–0.80) between factors of the PSS-10 and respective items loading on them.

### 3.3. Invariance of the PSS-10 across Different Groups

Table 3 shows that the PSS-10 seems to be generally fitting similarly across different groups of age, marital status, and monthly expenditure (configural invariance). In different groups, all items of the PSS-10 significantly loaded on the corresponding factors along with overall good fit on all indices except for RMSEA, which was a bit on the high side in both age groups and in married participants. Both ΔCFI and ΔRMSEA indicate equivalence of the 2-factor PSS-10 model in terms of factor variances, covariances, and errors between groups. Although χ^2^ was marginally significant (*p* = 0.047) for the structural covariances model across age groups, constraining all the paths to equality between groups would not cause a significant decrease in model fit. 

### 3.4. Concurrent Validity

Table 4 shows that the PSS-10 and its subscales demonstrate concurrent validity. As hypothesized, both the PSS-10 and PSS-stress positively correlated with the BDI-II and negatively correlated with life satisfaction. On the contrary, the PSS-coping negatively correlated with the BDI-II and positively correlated with life satisfaction.

### 3.5. Convergent Validity and Reliability

Subscales of the PSS-10 express a better convergent validity than the total scale, as highlighted by the higher values of item-total correlations for both subscales (Table 1). Likewise, alpha (α) coefficient of the positive and negative subscales (coping and distress, 0.79 and 0.86, respectively) of the PSS-10 was better than that of the total scale (α = 0.67), indicating good internal integrity of both subscales. As shown in Table 1, removal of item 4 and item 5 from the PSS-10 would slightly increase α to 0.69 and 0.71, respectively. On the other side, removal of any item from the positive or negative subscales would decrease the value of α.

## 4. Discussion

The present study evaluated the psychometric properties of the Arabic version of the PSS-10 in a sample of female university students. Both EFA and CFA revealed that the PSS-10 has a pure structure, which covers two distinct factors (coping and distress), and the best fit was obtained by allowing these factors to correlate with each other. Overall, the PSS-10 has expressed its validity as a measure of stress perception in this sample, and measurement invariance was consistent between different groups of age, marital status, and average monthly expenditure. Compared with the total PSS-10, its subscales demonstrated better convergent validity, internal consistency, and concurrent validity.

EFA revealed two distinct factors with clean and satisfactory item loadings on the corresponding factors. Likewise, CFA denoted that a correlated 2-factor solution describes the best fit of the data. Convergent validity tests lend further support to the 2-factor structure since subscales of the PSS-10 had considerably higher item-total correlations than the total PSS-10. Likewise, internal consistency of the subscales of the PSS-10 was considerably higher (α = 0.79 and 0.86) than that of the total scale, which was rather low (α = 0.67). In line with this, Perera et al. noted that the four positively stated items, which make up the coping subscale, can act as a nuisance variable that introduces unwanted variation when scoring the PSS-10 [21]. Despite the existing debate on the dimensionality of the PSS-10 [2], the greatest portion of the literature supports its 2-factor structure. This is evident in previous studies examining the dimensionality of the PSS-10 in student samples [7,8,22], in the general population [13], and in samples that encounter special types of distress such as military personnel (who suffer work-related conflicts and oppression) [12] as well as in patients with chronic and incurable disorders such systemic lupus erythematosus [11]. In general, our findings along with previous reports denote that although the PSS-10 is used as a measure of global stress, its structure is not dimensional given that its positively worded and negatively worded items cover two unique latent constructs.

Of interest, the PSS-10 holds measurement invariance among younger and older students, married and single students, as well as students with high and low average monthly expenditure. Inconsistent with these reports, some studies report that the PSS-10 does not work similarly across different groups, e.g., between athletes and non-athletes [7]. However, invariance is reported for strict factorial invariance [7]. Nonetheless, achieving strict invariance is rare, which prompts some researchers to consider its evaluation as unnecessary [2]. Because the current analysis is based on a subset of data from a large study examining eating patterns among female students and their association with emotional negativity, it was not possible to test for invariance across gender in our study. However, Denovan and colleagues reported equivalence of the loadings of a bifactor solution of the PSS-10 among male and female university students in the UK [2]. Altogether, the PSS-10 can be used effectively to reflect on stress perception among different groups.

Concurrent validity tests show that the PSS-10 possesses the ability to identify cases with high stress appraisal as it significantly correlated with scores of the BDI-II. However, the distress subscale may function better given that it correlated more strongly with depression scores and its negative correlation with life satisfaction was greater than that of the total PSS-10 scores (Table 4). This finding is consistent with results of convergent validity tests. In particular, positive items had remarkably low correlations with total scores of the PSS-10 (Table 1), and the removal of some of these items would result in considerable improvement of the internal consistency of the scale. On the other hand, the distress subscale demonstrates higher internal consistency and greater item-total correlations—the removal of any item would reduce its internal integrity. Overall, these findings denote that it may be necessary to consider scores of subscales of the PSS-10 along with its total score when screening for stress perception before and after treatment. Similar studies report significant positive association of the PSS-10 with measures of anxiety and depression (Patient Health Questionnaire (PHQ) and Generalized Anxiety Disorder 7-item scale (GAD-7)) among Chinese university students [8]. The PSS-10 and its subscales also positively correlate with negative emotions on the Positive and Negative Affectivity Schedule among students in the UK [2]. Overall, these results show that the PSS-10 is sensitive to aspects of distress, and thus, can be used to measure the improvement of mental status following medical treatment.

### Strengths and Limitations

This study, being the first to examine psychometric properties of the Arabic version of the PSS-10 through a sophisticated set of analyses (EFA, CFA, invariance, convergent validity, and concurrent validity), contributes significantly to the current literature. It highlights the importance of considering scores of subscales of the PSS-10 along with its total score when assessing stress perception. It also signifies that the PSS-10 can be used to consistently detect stress appraisal across different groups. On the other hand, recruiting only female participants is a major limitation, which made it impossible to evaluate invariance of the scale across gender groups in the current study. The sample was also derived from a single university, which entails uncertainty of attaining the same results in students from other universities. These limitations should be considered when interpreting the findings.

## 5. Conclusions

This study examined various psychometric properties of the Arabic version of the PSS-10. A correlated 2-factor structure of the scale has a merit of behaving similarly among different groups. The distress subscale can better screen for distress than the whole scale, which has implications for identifying students, mainly females, with high distress who need to be enrolled in specific intervention programs. It may also be sensitive to changes that follow such interventions. Therefore, it may reflect on the effect of treatments delivered to distressed students in support of efforts for promoting quality of life among university students.

## Figures and Tables

**Figure 1 brainsci-11-00419-f001:**
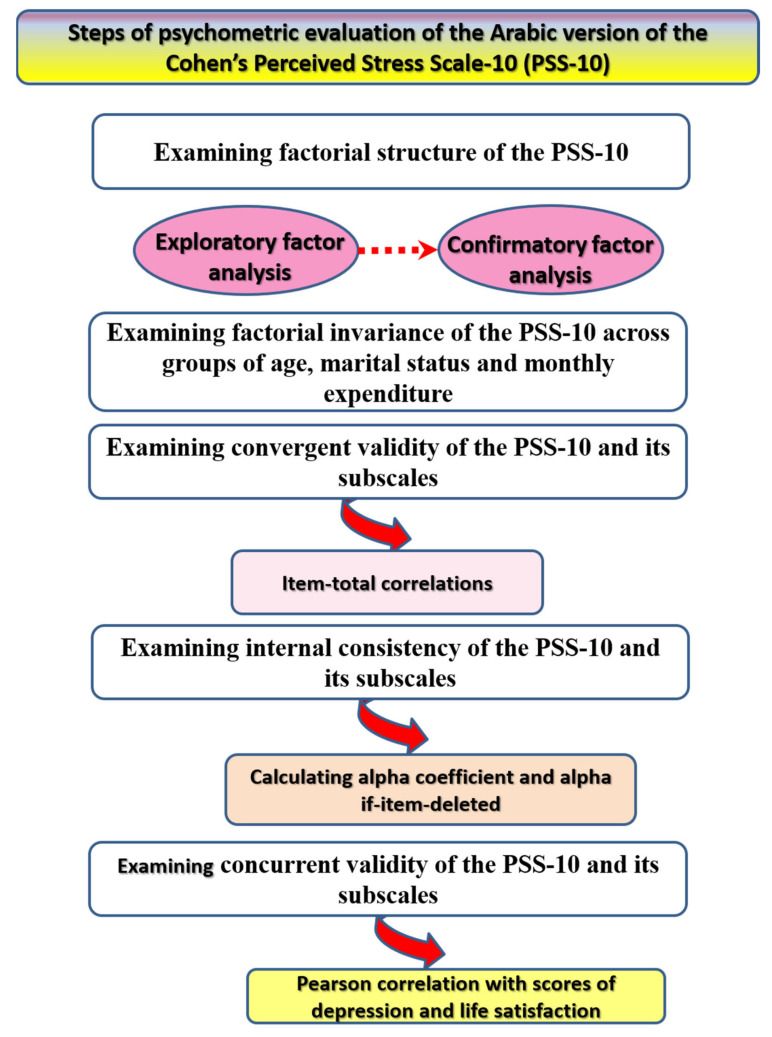
Planned methods for the psychometric evaluation of the Arabic version of the Cohen’s Perceived Stress Scale-10 (PSS-10).

**Figure 2 brainsci-11-00419-f002:**
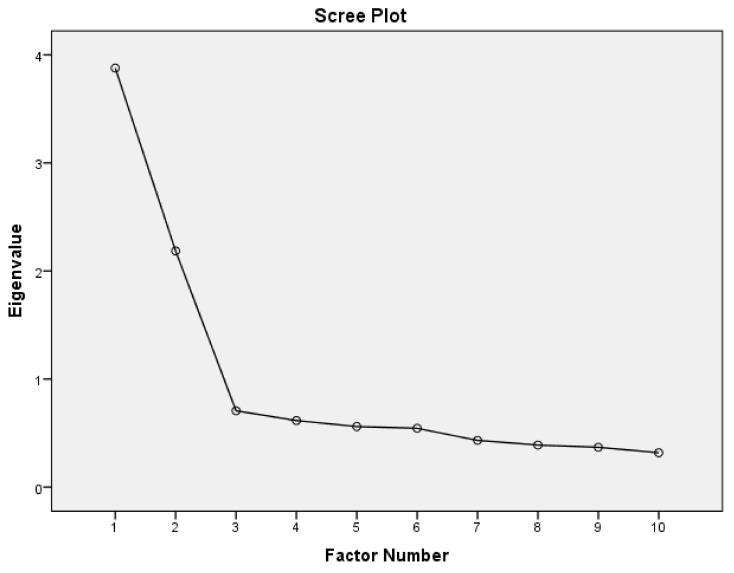
Scree plot of eigenvalues from exploratory factor analysis of the Arabic version of the Cohen’s Perceived Stress Scale-10 items.

**Figure 3 brainsci-11-00419-f003:**
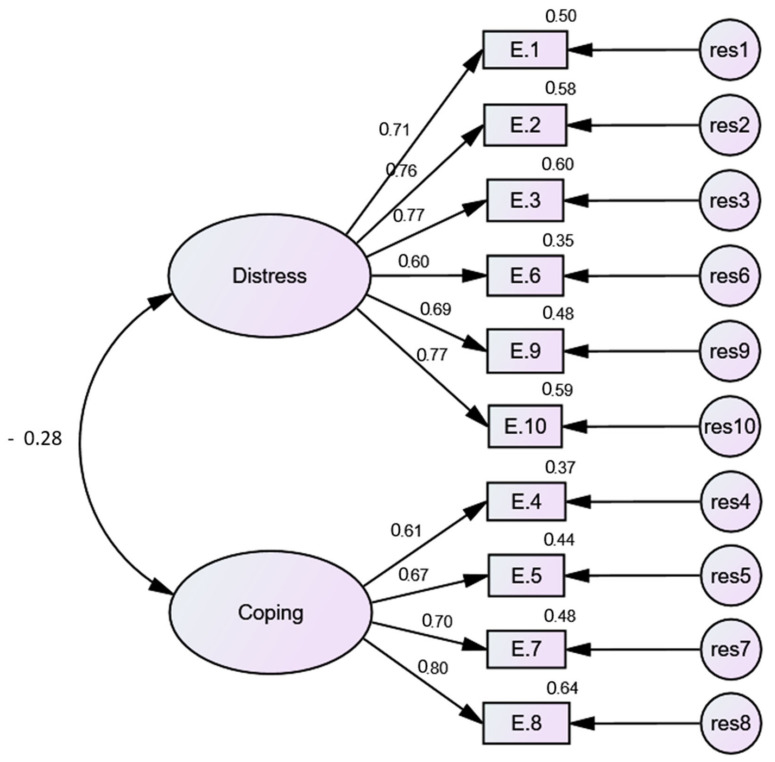
Correlated 2-factor structure of Cohen’s Perceived Stress Scale-10 (PSS-10).

**Table 1 brainsci-11-00419-t001:** Results of exploratory factor analysis of the Arabic version of the Cohen’s Perceived Stress Scale-10 items (extraction by maximum likelihood with Oblimin rotation and Kaiser normalization).

Item Number	Factor 1	Factor 2	Item-Total Correlations		
			PSS-10	Factor 1	Factor 2
1	0.706		0.498	0.654	
2	0.758		0.477	0.699	
3	0.775		0.517	0.709	
4			0.152		0.539
5		0.616	0.026		0.576
6	0.598	0.661	0.468	0.550	
7			0.165		0.593
8		0.702	0.096		0.665
9	0.700	0.799	0.527	0.637	
10	0.769		0.472	0.713	
Alpha coefficient			0.67	0.86	0.79
Range of alpha if-item-deleted			0.62–0.71	0.83–0.86	0.70–0.74

PSS-10: Cohen’s Perceived Stress Scale-10. After oblique rotation, the correlation between the two factors was –0.282.

**Table 2 brainsci-11-00419-t002:** Goodness-of-fit indices for tested models tested in confirmatory factor analysis.

Models	χ^2^	df	*p*	CFI	TLI	RMSEA	RMSEA 90% CI	SRMR
Model 1	107.014	35	0.000	0.948	0.934	0.074	0.058 to 0.090	0.0991
Model 2	86.43	34	0.000	0.962	0.950	0.067	0.047 to 0.081	0.0479

χ^2^: chi-square; df: degrees of freedom; CFI: comparative fit index; TLI: Tucker–Lewis index; RMSEA: root mean square error of approximation; CI: confidence interval; SRMR, standardized root mean residual.

**Table 3 brainsci-11-00419-t003:** Invariance of the 2-factor model of the Perceived Stress Scale-10 across different groups.

Groups	Models	χ^2^	df	*p*	Δχ^2^	Δdf	*p*(Δχ^2^)	CFI	ΔCFI	TLI	ΔTLI	RMSEA	ΔRMSEA
Age	20 years or less	80.476	34	0.000				0.941		0.929		0.077	
	Above 20 years	71.687	34	0.000				0.931		0.909		0.087	
	Configural	152.193	68	0.000				0.941		0.922		0.057	
	Metric	156.789	76	0.000	4.595	8	0.900	0.943	−0.002	0.933	−0.011	0.053	0.004
	Strong	164.754	79	0.000	7.966	3	0.047	0.940	0.003	0.933	0.001	0.054	−0.001
	Strict	172.83	89	0.000	8.076	10	0.621	0.941	−0.001	0.940	−0.007	0.050	0.004
Marital status	Married	48.644	34	0.000				0.914		0.886		0.106	
	Single	80.348	34	0.000				0.962		0.949		0.063	
	Configural	129.856	68	0.000				0.955		0.941		0.049	
	Metric	134.310	76	0.000	4.454	8	0.814	0.958	−0.003	0.950	−0.009	0.045	0.004
	Strong	135.486	79	0.000	1.176	3	0.759	0.959	−0.001	0.953	−0.003	0.044	0.001
	Strict	144.096	89	0.000	8.610	10	0.570	0.960	−0.001	0.960	−0.007	0.041	0.003
Monthly expenditure	20,000 ED ✺ or less	57.895	34	0.006				0.966		0.955		0.059	
	More than 20,000 ED	72.31	34	0.000				0.945		0.927		0.080	
	Configural	130.211	68	0.000				0.956		0.941		0.049	
	Metric	140.248	76	0.000	10.037	8	0.262	0.954	0.002	0.946	−0.005	0.047	0.002
	Strong	144.666	79	0.000	4.419	3	0.220	0.953	0.001	0.947	−0.001	0.047	0.000
	Strict	151.703	89	0.000	7.037	10	0.722	0.955	−0.002	0.955	−0.008	0.043	0.004

χ^2^: chi-square; df: degrees of freedom; CFI: comparative fit index; TLI: Tucker–Lewis index; RMSEA: root mean square error of approximation; ED: Emirati dirhams; ✺: the estimated equivalent in Euro and US dollar is 4564 and 5445, respectively.

**Table 4 brainsci-11-00419-t004:** Correlations between the Perceived Stress Scale-10 and its subscales with scores of depression and life satisfaction.

Variables	1	2	3	4	5
1. PSS-10	--				
2. Factor 1 (distress)	0.829 **	--			
3. Factor 2 (coping)	0.371 **	−0.211 **	--		
4. BDI-II	0.444 **	0.688 **	−0.372 **	--	
5. Life satisfaction	−0.254 **	−0.497 **	0.378 **	−0.672 **	--
Mean (SD)	21.5 (5.3)	13.3 (5.1)	8.2 (3.0)	14.7 (9.1)	7.1 (2.3)

PSS-10: Cohen’s Perceived Stress Scale-10, BDI-II: Beck Depression Inventory-II, **: *p* < 0.001.

## Data Availability

The data cannot be publicly shared because the dataset contains potentially identifying and sensitive participant information. Data will be made available upon request.

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
