# Peer review of "The Arabic Version of the Cohen Perceived Stress Scale: Factorial Validity and Measurement Invariance"

_brainsci, 2021, doi:10.3390/brainsci11040419_

Round 1
Reviewer 1 Report
Thank you for the opportunity to review the manuscript entitled “The Arabic version of the Cohen perceived stress scale: factorial validity and measurement invariance” (brainsci-1127487).
The authors conducted an exploratory (EFA) and a confirmatory factor analysis (CFA) on the Arabic version of the PSS-10 in cross-sectional data of 379 female Emeriti students. In addition, internal consistency (Cronbach’s alpha), convergent validity (item-total correlation), and concurrent validity (association with depression and life satisfaction), and construct validity (correlation with depression, and life satisfaction) were tested. In general, the PSS-10 appears to be a valid tool also in the United Arab Emirates. The manuscript is well written and the analyses are appropriate. I only have very few recommendations for improvement.
The abbreviation PSS-10 should be introduced in the abstract.
In line 55, it is mentioned that stress is especially relevant for students below the age of 26 years. In this study the age categories were formed at the age of 20 years, why?
What is the difference between convergent and concurrent validity?
The sample consists of female university students of one university. The generalisation remains uncertain and should therefore be discussed.
The internal consistency was rather low, especially for the overall scale. The authors might discuss also this point.
In Table 4, it would be helpful for the reader if the numbers also appear in the first column.
I hope my recommendations and comments help to improve the paper.
Author Response
Manuscript ID: brainsci-1127487.
Title: The Arabic version of the Cohen perceived stress scale: factorial validity and measurement invariance
Response to Comments of Reviewer 1
We appreciate the Reviewer’s helpful comments and concern for clarity as indicated by the provided comments. The comments are addressed line-by-line as shown below. Replies come underneath in red.
The abbreviation PSS-10 should be introduced in the abstract.
Authors’ response: Yes, we have introduced the abbreviation PSS-10 in the abstract after its first definition (line 19).
In line 55, it is mentioned that stress is especially relevant for students below the age of 26 years. In this study the age categories were formed at the age of 20 years, why?
Authors’ response: Thank you for such a sharp and important observation. The reported age in line 55 is based on large samples with a wide range, which is different in the current study. Because some CFA measures are affected by sample size (e.g., chi square), we grouped students according to their median age (age was not normally distributed) to ensure that the sub- group analysis is based on groups relatively equal in size.
What is the difference between convergent and concurrent validity?
Authors’ response: Although these constructs are closely-related and are used interchangeably, they are different. We have provided definitions for both terms in the text (lines 128-138).
The sample consists of female university students of one university. The generalisation remains uncertain and should therefore be discussed.
Authors’ response: Yes, we have modified the limitations section to include the uncertainty of results replication among students from other universities.
The internal consistency was rather low, especially for the overall scale. The authors might discuss also this point.
Authors’ response: Thank you very much. Yes, we have included low reliability of the whole scale in the text supporting the use of the distress subscale (second paragraph in the Discussion).
In Table 4, it would be helpful for the reader if the numbers also appear in the first column.
Authors’ response: Yes, we have added the numbers in the first column.
We hope that the comments were properly handled and that the revised version will be suitable for publication.
Best regards,
Corresponding author
Reviewer 2 Report
In the manuscript entitled “The Arabic version of the Cohen perceived stress scale: factorial validity and measurement invariance”, Amira Mohammed Ali and colleagues evaluated the psychometric properties of the Cohen Percieved Stress Scale (10 items) in 379 female Emeriti students. Because stress is a significant factor contributing to mental disorders, developing or re-assessing different diagnostic scales in order to optimize their application in various populations has a potential clinical value.
The study is well designed and performed. I have only minor comments:
- A rationale for performing the study only on female student participants should be provided
- In description to Table1, information about factor 1 and factor 2 is missing
- In Results’ 3.1. subsection and Table3, the Authors provide currency Emirati dirhams, please provide an estimated equivalent in Euro
- In the Conclusions, I miss the clear summary of the advantages of using the modified scale/subscale compared to complete PSS-10 scale - when and in whom the “new” scale could be used?
Author Response
Manuscript ID: brainsci-1127487.
Title: The Arabic version of the Cohen perceived stress scale: factorial validity and measurement invariance
Response to Comments of Reviewer 2
We appreciate the Reviewer’s helpful comments and concern for clarity as indicated by the provided comments. The comments are addressed line-by-line as shown below. Replies come underneath in red.
- A rationale for performing the study only on female student participants should be provided
Authors’ response: Yes, the analysis included a subset of data from a large study examining eating patterns among female students and their association with emotional negativity. We have noted that in the text (page 11, first paragraph).
- In description to Table1, information about factor 1 and factor 2 is missing
Authors’ response: Yes, we have added some information about factor 1 and factor 2 in description to Table1 (lines 233-237).
- In Results’ 3.1. subsection and Table3, the Authors provide currency Emirati dirhams, please provide an estimated equivalent in Euro.
Authors’ response: We heartfully admit that these values may not be easy to interpret for readers from non-Arab countries. However, this scale will be always used for Arab subjects and the information within the Arabic context is important since Euro is not commonly used in these countries. To tackle this issue, we have reported on the estimated equivalent in Euro and US dollars in the footnote of Table 3.
- In the Conclusions, I miss the clear summary of the advantages of using the modified scale/subscale compared to complete PSS-10 scale - when and in whom the “new” scale could be used?
Authors’ response: Based on the fact that the study has recruited only a female sample, we have specified in the conclusion that the scale, particularly the distress subscale, works better among female students with high distress to be enrolled in interventions designed to promote emotional wellness and following treatment to assess its effectiveness. In the Conclusions, the sentence “The distress subscale can better screen for distress than the whole scale, which has implication for identifying students, mainly females, with high distress who need to be enrolled in specific intervention programs…” has answers to the question of the reviewer.
We hope that the comments were properly handled and that the revised version will be suitable for publication.
Best regards,
Corresponding author